# The Use of Wearable Sensor Technology to Detect Shock Impacts in Sports and Occupational Settings: A Scoping Review

**DOI:** 10.3390/s21154962

**Published:** 2021-07-21

**Authors:** Ingrid Eitzen, Julie Renberg, Hilde Færevik

**Affiliations:** 1Department of Smart Sensor Systems, SINTEF Digital, 0373 Oslo, Norway; 2Department of Health Research, SINTEF Digital, 7034 Trondheim, Norway; julie.renberg@sintef.no (J.R.); hilde.ferevik@sintef.no (H.F.)

**Keywords:** wearable sensor technology, inertial measurement unit, optimal load, sports, occupational health, foot strike, landing impacts, head impacts

## Abstract

Shock impacts during activity may cause damage to the joints, muscles, bones, or inner organs. To define thresholds for tolerable impacts, there is a need for methods that can accurately monitor shock impacts in real-life settings. Therefore, the main aim of this scoping review was to present an overview of existing methods for assessments of shock impacts using wearable sensor technology within two domains: sports and occupational settings. Online databases were used to identify papers published in 2010–2020, from which we selected 34 papers that used wearable sensor technology to measure shock impacts. No studies were found on occupational settings. For the sports domain, accelerometry was the dominant type of wearable sensor technology utilized, interpreting peak acceleration as a proxy for impact. Of the included studies, 28 assessed foot strike in running, head impacts in invasion and team sports, or different forms of jump landings or plyometric movements. The included studies revealed a lack of consensus regarding sensor placement and interpretation of the results. Furthermore, the identified high proportion of validation studies support previous concerns that wearable sensors at present are inadequate as a stand-alone method for valid and accurate data on shock impacts in the field.

## 1. Introduction

Every load applied to the body will lead to an impact force on the joints and the articular cartilage, to which the body must respond with shock attenuation. This is not necessarily harmful—it is well known that loading is a prerequisite for good joint health, as mechanical forces contribute to maintain the specialized function of articular cartilage and subchondral bone [1]. However, at the same time, excessive strain, either over time or as a result of a single impact with abnormal stress, may cause damage to the joint, which in the short term can lead to pain and reduced function and in the long term to the development of osteoarthritis [2,3,4]. Furthermore, the body can be exposed to shock impacts that may hurt other structures—muscles, bones or inner organs, and of particular concern, the brain. To identify and define the threshold for shock impacts that may cause damage will be of utmost relevance to define tolerable training and activity loads and to prevent both acute and stress-related injuries. However, to explore this, we first need to gain a better and more in-depth understanding of which methods can monitor shock impacts most accurately in real-life settings in different contexts.

Wearable sensor technology provides an alternative to classical laboratory-based assessments of human performance that enables real-time monitoring in natural environments [5], without the cumbersome set-up procedure and limitations related to space [6]. Assessments of human motion by wearable sensor technology generally utilize Inertial Measurement Units (IMUs), which usually consist of triaxial accelerometers, often complemented by three axis gyroscopes and/or three axis magnetometers [7]. The rapid growth in commercially available wearable sensors makes them easily accessible and affordable [8]. In this study, we will use a scoping review methodology to map the use of wearable sensor technology to detect shock impacts in two domains: sports and occupational settings.

### 1.1. Sports

Wearable sensor technology in sports has been reported to address injury prevention, motion analysis, technique classification, and performance and skill-level assessments [9]. Existing studies have addressed a wide audience, spanning from youth sports and recreational activity to top elite athletes [8,9,10]. In the past years, numerous studies have also included IMUs in combination with global positioning systems (GPS) and heart-rate monitoring to investigate various training paradigms in different sports and assess the balance between improvements in athletic performance and risk of overtraining and potential injuries [11,12]. Training load or stress is within this context typically defined as the sum of external (training volume and intensity) and internal (physiological parameters and perceived exertion) loads combined [12,13,14]. Many of these studies have used wearable sensors to monitor the total exposure over time—during a training session, a period of training or a whole season, most often in team sports [15,16,17,18], and in running [14,19,20]. However, within the sports domain, numerous types of sports and activities involve shock impacts that may be harmful from just one or a limited number of impacts—alpine skiing, running, gymnastics, team sports such as soccer, volleyball, and different types of invasion sports such as rugby and football, to mention a few. In this study, we will focus specifically on the use of wearable sensors for detecting shock impacts at these events.

### 1.2. Occupational Settings

Wearable sensor technology enables assessments of real-time measurements of workers in their natural setting. Spook et al. [21] recently investigated worker needs and preferences with regard to implementation of sensor technology to measure and monitor physical job demands and work exposures, and they found that workers were positive to such methods and could see the value of both real-time feedback and access to data on demand. However, a recent narrative review from Lim and D’Souza [22] states that despite growing interest, research utilizing wearable sensor technology in applied occupational settings is still sparse. Their findings reveal that the use of inertial sensors for biomechanical exposure assessment in occupational settings has gained increased attention the past few years, addressing different contexts including healthcare facilities, shift workers in different physically demanding jobs, as well as white-collar work. However, it may seem as though quantification and descriptions of postural information have been the primary scope in existing studies. Awkward postures are regarded as a determinant for the development of musculoskeletal disorders and is therefore an important measure to quantify in occupational settings. Still, depending on the occupational group in question, objective measurements of other exposures may also be important. As in the sports domain, some workers are repeatedly exposed to shock impacts. Construction workers using tools with high single accelerations, such as nutrunners, is one example. The operation of high-speed boats is another. Reduced physical performance following a large number of high mechanical shock impacts during a high-speed boat transit has been shown [23]. The suggested mechanisms behind the degradation on performance were muscle damage and localized muscle fatigue due to repeated eccentric contractions to attenuate the shock impacts. Furthermore, high-speed boat crew members often experience acute traumas and chronic musculoskeletal pain related to the nautical activity [24,25,26]. In some instances, shock impacts can be measured on the objects, as in the high-speed boats. However, as shock waves are attenuated by biological tissues within the body, the effect of the shock impacts on different anatomical structures can be very different from that measured on the boat itself. Therefore, in this study, we are looking into whether there are existing methods to objectively measure shock impacts using wearable sensors in occupational settings.

### 1.3. Study Aim

It appears that research utilizing wearable sensors both within the sports domain and in occupational settings primarily is recognizing the value of monitoring total exposure, risk factors related to activity, overloading or ergonomics, and resultant consequences (such as fatigue or injuries), rather than the specific events that may cause harm. Furthermore, it seems as though the utilization of wearable sensors for specific events has come somewhat further in the sports domain. There is a need to examine whether this impression is true, or if current literature also presents reports on the use of wearable sensor technology to detect undesirable external stimuli in the form of potentially harmful shock impacts in sports and occupational settings. To come further in defining thresholds for whether joint loads are beneficial or damaging in different contexts, objective data from real-life settings quantifying single shock impacts are warranted as a supplement to existing methods that monitor total load over time or exposures to ergonomic strain. Therefore, the main aim of this scoping review is to present an overview of existing methods for assessments of shock impacts using wearable sensor technology within sports and occupational settings. The following research questions will be addressed:What type of wearable sensor technology is used to measure shock impacts?In what types of activities is wearable sensor technology used to measure shock impacts?Which sensor placements and outcome measures are used when measuring shock impacts using wearable sensor technology?Which knowledge gaps are apparent in the literature regarding wearable sensor technology for assessments of shock impacts within sports and occupational settings?

## 2. Materials and Methods

Scoping reviews serve to synthesize evidence and assess the scope of literature on a given topic. In contrast to a systematic review, a scoping review does not aim to synthesize evidence from the included papers in order to evaluate study quality or provide evidence to inform practice. Rather, the main purpose is to map the available evidence.

### 2.1. Protocol

This scoping review follows the scoping review methodological framework to systematically map a research area, identify the main sources and types of available evidence, and identify research gaps in the existing literature [27]. The scoping review protocol was drafted using the PRISMA methodology and its extensions for scoping reviews (PRISMA-ScR) [28]. Results are reported in line with Moher et al. [29].

The authors responsible for this study compose a cross-scientific team with expertise in physiology, sports medicine, biomechanics, movement analysis and wearable sensor technology. The study protocol was established from discussions on the overall aim, selected fields to be included, and the context and limitations of the research questions to be addressed. From this, the team agreed on search terms, inclusion and exclusion criteria, search strategy, and selection of databases to search.

### 2.2. Eligibility Criteria

To be included in the data material for this scoping review, papers had to measure shock impacts on the whole body or specific body regions with the use of wearable sensor technology. Wearable sensors were limited to either sensors attached to the body, to clothing, or to equipment used by the individual. Furthermore, the measurement methods needed to be relevant in a sports or occupational setting. Full-text review articles were retrieved if they included relevant measurement methods of shock impacts, but they were not included in the data material. Peer-reviewed journal papers or conference papers were included if they accommodated the initial criteria, were written in English, and were published between 2010 and 2020. Due to the rapid development of this field, a 10-year limitation was chosen to narrow the results to relevant sensor technologies likely still in use, with the rationale that technology developed before 2010 still in use would appear in publications within the chosen time frame. Papers were excluded if they did not fit the purpose of the study, the included subjects were patients or individuals with a functional impairment, disability, or illness, or if they focused on movement quantification, total load, movement quality, or technique only. Regular gait analysis was deemed not relevant for the purpose of this paper. Papers on gait or walking were eligible for inclusion only if related to a specific work-related context and if they specifically reported measurement of impact forces.

### 2.3. Information Sources

To identify possibly relevant publications, on 7 September 2020, the following bibliographic databases were searched: MEDLINE, SportDiscus, Scopus, PubMed, Compendex, and ISI Web of Science. The search strategies were drafted by the help of a librarian (Trude Eikebrokk) and further refined through discussions between the three authors. The final search strategy for MEDLINE can be found in Appendix A, Figure A1. The final search results were exported into EndNote and then into the software tool Covidence [30]. Duplicates were removed in Covidence.

### 2.4. Selection of Sources of Evidence

To ensure consistency among reviewers, all three authors performed a title and abstract screening calibration for a random selection of 25 papers. The results were compared and led to minor adjustments of the screening tool. Then, the process was repeated with another 25 randomly selected papers before the screening tool was amended once more to assure consistency between reviewers before the screening of all abstracts started. The final screening tool for the title and abstract screening and the full text screening can be found in Appendix B, Table A1 and Table A2. The software tool Covidence was used for extraction of relevant titles and abstracts, and then for the extraction of eligible papers. First, the title and abstracts, and thereafter the full text publications of all included abstracts, were screened by two reviewers to identify papers for final inclusion in the data material. Disagreements on study selection were resolved by discussion among all three reviewers, first when screening titles and abstracts, and later when reading all eligible full-text papers.

### 2.5. Data Charting Process

A data-charting form was developed by the three reviewers to determine which variables to extract from the eligible studies. One reviewer was in charge of charting the data and consulted the other two whenever there was any doubt as to what data should be extracted. Then, the results were discussed.

### 2.6. Data Items and Synthesis of Results

The type of data extracted from the eligible studies were article characteristics (e.g., country of origin, study design, aim or purpose), article methods (main outcome measurement of relevance to this review, population used, type of sports, impact site), types of wearable sensor technology used, additional technology used for validation, and the key findings of relevance. We grouped the studies by types of activities investigated. The data chart can be found in Table 1.

The country of origin, study design, participants, type of wearable sensor technology used, sensor placement, and additional technology for validation were extracted and counted (Table 2).

## 3. Results

The initial search resulted in 628 abstracts that were screened by two independent reviewers. Of these, 543 were not accommodating the inclusion criteria. Eighty-five publications were selected for full-text screening. Following the reading of these papers, 51 additional papers were excluded, leading to a final number of 34 included papers in the data material (Figure 1).

No papers accommodating the study criteria for final inclusion reporting measurement methods for shock impacts in occupational settings were identified. To assess whether the search results in the occupational domain were due to a too narrow search strategy, we performed a new search on 21 October 2020 in the same databases and included “military”, “personnel”, “police”, “speed boat*”, and “construction” in the search string. This did not result in any additional eligible studies. Thus, the final extraction includes papers in the sports domain only. Of the 34 full-text papers included from the sports domain, 33 were peer-reviewed journal articles [31,32,33,34,35,36,37,38,39,40,41,42,43,44,45,46,47,48,49,50,51,52,53,54,55,56,57,58,59,60,61,62,63], and one was a conference abstract [64]. The included papers are listed in Table 1 with the following information included: author(s), year of publication, type of sport, study design, study aims, main outcome measure, population, type(s) of wearable sensor technology and placement of sensor, additional technology for validation, and key findings. With regard to the main outcome and key findings, the information extracted in Table 1 is limited to the outcome of relevance and key findings of relevance for the purpose of this scoping review and the defined research questions. An excerpt of the demographic and technological characteristics of the included studies is given in Table 2.

The majority of the included papers presented wearable sensor measurements of landing impacts during foot strike in running (number of studies included = 12), head impacts in invasion and team sports (soccer, football, rugby, and lacrosse) (number of studies included = 10), and impacts in different jump landings on feet, including somersaults (number of studies included = 6). One of the studies classified as measuring impact during jump landings on feet also included landing on hands in gymnastics. The remaining included papers presented the following other main outcomes: breaking and stopping in running (*n* = 1), whole body impact in rugby (*n* = 1), landing, cutting, and stepping in badminton (*n* = 1), GRF predicted from trunk-mounted sensor in running (*n* = 1), change in mechanics during running derived from peak COM acceleration (*n* = 1), and foot strike during multiple motions/plyometrics (*n* = 1).

## 4. Discussion

The main aim of this scoping review was to present an overview of existent methods using wearable sensor technology for assessments of shock impacts within sports and occupational settings. We specified four research questions:What type of wearable sensor technology is used to measure shock impacts?In what types of activities is wearable sensor technology used to measure shock impacts?Which sensor placements and outcome measures are used when measuring shock impacts using wearable sensor technology?Which knowledge gaps are apparent in the literature regarding wearable sensor technology for assessments of shock impacts within sports and occupational settings?

Surprisingly, no papers accommodating the criteria for final inclusion were identified in the occupational domain. Thus, the lack of studies assessing shock impacts using wearable sensor technology within occupational settings may be stated to be the overall biggest knowledge gap identified in this study.

High-speed boats personnel and some types of construction workers are examples of workers known to be exposed to repeated shock impacts, which are believed to negatively affect health and performance. For industry workers, exemplified from the petroleum industry, walking on hard surfaces and climbing stairs for long shifts, five days a week, year round wearing heavy and stiff safety shoes is believed to cause musculoskeletal disorders in the lower back and lower limbs [65]. In work environments such as these, it is the high repetitions, the long durations, and also the accumulated load of different unwanted exposures that pose a risk [66]. As stated previously, quantification and descriptions of postural information has been the primary scope in existing studies within the occupational domain. However, if one wants to establish associations between exposures and health and performance outcomes, it is important to include the measurement of several hazardous exposures to account for the accumulated occupational load. On some occasions, relevant measures will include the dose and frequency of shock impacts during work. Thus, further research into the assessments of shock impacts using wearable sensor technology within occupational settings seems warranted.

This scoping review indicates that the urge to find technology for assessing shock impact has been higher in the sports domain than in the occupational health area. The use of wearable technology in sports is driven by the need to understand biomechanics to prevent injury and to provide immediate feedback to athletes and coaches, which is a common interest for both. Although the benefits of wearable sensors systems (e.g., shock impact assessment) have significant potential for health risk mitigation in the occupational health domain, their ability to capture specific information that may be considered personal and private by the worker may hinder its utilization [67]. Hence, the potential conflict of interest by employers and workers may possibly explain the lack of research in this area as well as the lack of papers on shock impacts in the occupational domain. Another argument is that occupational health assessment is driven by standards and methods for risk assessment to prevent accidents from happening, that employers and workers do not see the usefulness of these functions in enhancing workers’ safety and health. Furthermore, the use of new methods for assessment has a much longer timeline for implementation and is still immature for many occupational health workers and their management.

Due to the absence of studies on occupational settings, the remaining part of this discussion will be concentrated on addressing each of the four research questions for the sports domain.

### 4.1. What Type of Wearable Sensor Technology Is Used to Measure Shock Impacts in Sports?

All the included studies in this scoping review included some kind of accelerometer, except for four studies that used insoles with force sensors. The type of accelerometer varies, from uni- to triaxial, and it also varies whether the accelerometers were embedded in an IMU that also included a gyroscope and a magnetometer. All studies had a description of the type of sensors used; however, the level of detail was somewhat inconsistent, making it challenging to compare differences among the different types of accelerometers used. More than 20 different types or brands of sensors were utilized in the studies, but few studies had any line of argument on why they had chosen this particular brand or type. The majority of the studies provided a description of the number of sensors used and sensor placement, allowing others to replicate the study protocol. However, the level of detail varied also on this aspect.

With regard to which type of data most often is extracted from the accelerometer/IMUs, peak acceleration is the dominant outcome measure. This applies to studies assessing running (foot strike), different landings, as well as head impacts. For studies utilizing insoles, force and pressure distribution is the outcome of interest. A detailed discussion on the outcome measures considered in the different contexts will be provided later in this discussion.

Of the 34 studies included, as many as 25 utilized additional technologies to the wearable sensors. For studies on running, seven studies used force plates or instrumented treadmills with force plates embedded [34,37,44,47,49,57,63], two used force plates and camera recordings [55,60] and three used camera recordings only [38,52,59]. For assessments of landing impacts, force plate data was reported in four studies [31,33,62,64], force plate and camera recordings were reported in one study [32] and camera recordings only were reported in one study [46]. The main reason for adding these measurement technologies was that the studies had a single intervention design, with validation of the data collected from the wearable sensor technologies as part of their main purpose. For studies concerning head impacts in invasion and team sports, camera recordings were added in seven of the included studies [36,40,41,48,53,54,58]. The combined use of wearable technology and other technologies, in many cases representing current gold standards, will be discussed in detail in the last section of this paper where identified knowledge gaps are addressed.

### 4.2. In What Types of Activities Is Wearable Sensor Technology Used to Measure Shock Impacts in Sports?

Two areas were identified to stand out regarding this question: peak impact during foot strike in running and head impacts in invasion and team sports such as football, lacrosse, rugby, and soccer. Thereafter, landings after jumps or plyometric activities seem to be given some attention in activities such as gymnastics, ballet, and non-invasive ball sports such as badminton.

Outdoor team sports is one of the areas were GPS early on was utilized as a method for monitoring player movements [68]. In addition, the use of Micro-Electro-Mechanical Systems (MEMS) technology including different types of IMUs and magnetic sensors in this area has expanded, allowing further assessments of total workload, technique, and physical performance [6,69]. However, it became apparent from this review that single shock impacts—other than head impacts—are not presented in the literature, neither for contact team sports such as soccer, football, or team handball nor other team sports such as volleyball and basketball. This may be somewhat surprising, given that several of these sports are characterized by jump landings, rapid changes of direction during high velocities, running, and pivoting activities, which all cause considerable loads on the lower extremities. It is further well known that athletes in these sports are exposed to considerable risk of injury to the lower extremities [70,71,72,73], both acute injuries such as meniscus and ligament tears, cartilage lesions, and fractures, and overuse injuries such as degenerative meniscal tears and jumper’s knee, to mention a few. For the latter case of overuse injuries, it can well be argued that monitoring total load and/or exposure over time may be the most relevant measure. This is also the impression from the current research literature. Even though the search strategy for this scoping review did not include “total load” or “movement quantification” as terms, we still identified and excluded 21 studies that had total load or movement quantification as their primary outcome, and numerous more studies on this topic can be found. Studies assessing total load will typically emphasize the number of jumps, jump height, total distance covered in running, velocity, and/or changes of direction during a game, a season, or a defined period of training and competition [68,74]. Differences in total load exposure between athletes with different positions on the field are also commonly reported [75,76,77], and the literature further reflect the inclusion of athletes at different ages and levels of performance. However, even though such data can be valuable for monitoring exposure, helping balance total training load, and developing strategies to prevent injuries, it could still be suggested that more insight into the effect of each shock impact event could be of interest. For example, the joint load exposure of a total number of jumps performed during a volleyball season will be strongly dependent on the forces that need to be absorbed during landing. Jump height can give an indication about the forces absorbed during landing. However, to identify single situations that imply such high loads that they present particular risk for either an acute injury or a rapidly developing overuse injury, specific assessments of shock impacts are required. An interesting observation in addition to the identified lack of studies on shock impacts in team sports including jumping is that no studies accommodating the inclusion criteria were found for track and field. This is particularly noteworthy for the high jump, the long jump, and the triple jump. These events imply high impact forces, especially in the last one-legged plant step before take-off. Measurements on the magnitude of the single shock impacts during this critical phase of the jump would probably be interesting both for assessments of performance and for injury prevention/load management in training.

In contrast to team sports, it is interesting to note that the approach seems different in studies on running. Even though total exposure (distance covered, surface, shoes) and running technique (forefoot, midfoot, or rearfoot landings, stride length) are topics under investigation in many studies on running, attention is also explicitly directed on single landing impacts: 15 studies on running [34,35,37,38,43,44,47,49,50,52,55,57,59,60,63] accommodating the inclusion criteria were identified. From these studies, knowledge on foot strike impacts can partly be transferred to several of the mentioned team sports. However, this will only be valuable for quite straightforward running—not changes of direction, breaking, stopping and pivoting, which are the movements causing the most strain on the joints and thus have the highest potential for causing damage.

Only one study was identified and included from winter sports: assessing drop landings in snowboard [64]. However, this was a conference abstract with information missing on design, primary outcome, and participants. The almost complete lack of studies investigating shock impacts during demanding winter sports came as a surprise. Several winter sports are conducted under harsh conditions, with considerable variation in the external demands posed by the environment [78]. As with contact team sports, the injury risk is high in winter sports such as alpine skiing, freeskiing, and snowboard [19,79,80]. Research on these sports relevant for sport-specific demands and injury prevention has been done under controlled laboratory settings. However, this does not necessarily transfer to real-life settings. The use of wearables in field studies in these sports seem to be concentrated on performance assessment and enhancement with evaluation of technique (for example turn switch detection) and tactical dispositions [78,81,82]. Another winter sport with considerable landing impacts is ski jumping. No studies on shock impacts during landings in ski jumping were identified for inclusion in this review. However, there are studies combining IMUs with force insoles, with attention toward the positioning of the skis and the landing technique [83,84]. The lack of studies investigating shock impacts during real-life performance in demanding winter sports is identified as a knowledge gap.

### 4.3. Which Sensor Placements and Outcome Measures Are Used When Measuring Shock Impacts Using Wearable Sensor Technology in Sports?

The rapid growth in commercially available wearable sensors makes them easily accessible and affordable. However, for the results to be valid and reliable, the methods used for placement and data analysis are central. In the included studies, the dominant main outcome measure is derived from acceleration. There seems to be consensus that peak tibial acceleration (also referred to as tibial shock) can be used as a proxy for the impact forces experienced at the tibia and thus the vertical impact loading in running [85], as it has been shown to correlate strongly with vertical loading rates [86]. Comparably, for head impacts, peak linear acceleration is interpreted as detected head impacts. Many studies use a defined threshold for what would be interpreted as a head impact, most often set as ≥10 g [41,42,51,54,58], but in three studies, it was even higher: ≥15 g [45,48] and ≥20 g [36]. Studies utilizing insoles base their outcomes on various measurements of force, vertical ground reaction force (vGRF), force impulse, and force/load distribution. Deceleration and braking are also reported as outcomes derived from both accelerometers and insoles.

In a recent review concerning measurements of tibial acceleration during running, Sheerin et al. [85] point out that different placements of accelerometers do not necessarily give comparable results. Furthermore, whether triaxial or uniaxial accelerometers are used will be of relevance, as accurate measurements from an uniaxial accelerometer is dependent on precise alignment along the long axis of the tibia [87]. Acceleration of the tibia occurs in three dimensions: axial, anteroposterior, and mediolateral. Triaxial systems will incorporate this and are thus less vulnerable to misinterpretations of the actual anterior acceleration of the tibia. In this review, six of the studies investigating foot strike in running with IMUs utilized triaxial accelerometers [34,35,37,52,57,60], two studies with the same first author used a combination of uni- and biaxial [43,44], whereas one study used uniaxial [47]. Furthermore, three studies utilized triaxial accelerometers, but with other primary outcomes than foot strike, and with accelerometers mounted on the trunk and/or several locations on the body [38,49,88]. With regard to the placement of those assessing foot strike as their primary outcome, there was considerable variation. Three studies had sensors placed at the anteromedial aspect of tibia [37,52,57], three studies had a combined placement on the heel counter of the shoe and anteromedial tibia [34,43,44], and the remaining three studies had sensors placed on the heel counter of shoe [35], on top of the shoe above the third metatarsal [47], and finally using a Lycra suit with multiple sensors on upper and lower extremities, trunk, and head [60]. This demonstrates a lack of consensus on sensor placement in the literature, which can make comparisons between studies very challenging. From the already mentioned review, Sheerin et al. also state that tibial acceleration measured by distally attached sensors gives higher values, which is a notion that is supported by Blazey et al. [87], who conclude that current evidence suggest IMU devices should be placed and fixated on the distal tibia. It should finally be noted that bone-mounted accelerometers have been shown to have the highest association with vertical load rates from force plates, with correlations of r = 0.97. In comparison, skin-mounted accelerometers, which are the relevant wearable for field situations and thus for the studies included in this material, have correlations of r = 0.70 [52].

With regard to head impacts, placement of the wearable sensors may be challenging, as they are prone to be broken by the impact. Wearable sensors for assessments of head impacts are available in several systems; they are embedded in instrumented helmets, headbands, mouthguards, and skin patches [89,90]. Of the ten studies included in the data material, skin patches (xPatch) were used in four studies [36,41,54,58], instrumented helmets (GForce Tracker and SpeedFlex/HIT) were used in five studies [36,40,42,45,51], a headband (SIM-G) was used in one study [48], and finally MVTrak, with a sensor placed in the ear canal, was used in one study [53] (number summarizes to eleven, as the study by Cortes et al. [36] utilized xPatch for females and GForce Tracker for males). Regardless of system, it is important to notice that the accelerations captured represent a combination of true head impacts—such as collisions or hitting the ground—and false detected impacts stemming from movement [39,41,91], e.g., change of direction, jumping, and decelerations. Even though impacts above 10 g, which is the most common threshold reported in the studies included in this review, are likely to be accrued by true impacts, several of the included studies show that this cannot be trusted unless confirmed by video. Linear and rotational acceleration magnitudes from lab studies have been shown to be over-predicted for sensor solutions in both skin patches and instrumented helmets [92]. False positive high acceleration impacts have further been revealed in field studies, and the importance of video confirmation of sensor-recorded events to remove false positives is in a recent systematic review by Patton et al. emphasized to avoid overestimation of head impact exposure [90]. Still, two-thirds of the included studies in their review did not include video. In this scoping review, numbers were higher, as seven out of ten included studies did include video confirmation, and several also emphasized the importance of doing so. Interestingly, Carey et al. [91] found the vast majority of high acceleration impacts, when defined as above 20 g, to be verified by video. This underlines the ambiguity of using 10 g as a threshold for head impacts, especially if not adding video to confirm events. The systematic review by Patton et al. [90] finally also states that even though the majority (74%) of the included studies used filtering algorithms, these remain inadequate. Thus, the studies included in this scoping review, as well as previous reviews, suggest that trusting wearable sensors as the only data source for assessments of head impacts in sports could imply an overestimation of potential harmful impacts, both in number and in severity.

Ground reaction force is an obvious variable of interest for mechanical analysis of eventual risk factors for injury, as well as for assessments of lower extremity load attenuation during sport activities that entail running, jumping, and landing. In this review, four studies utilizing pressure sensors were included. Of these, one study [59] used the Novel Pedar-X system to explore whether running speed affected plantar loads and to compare rearfoot versus mid- and forefoot strikers, without validating against a gold standard. The three other studies all validated against force plates [56,63] or an instrumented treadmill [55]. Stoggl and Martiner [56] validated the OpenGo sensor insole towards both a force plate and the PedarX assessing multiple motions, and found force impulses to be between 13 and 34% lower with OpenGo compared to force plates, emphasizing problems with force impact accuracy during short ground contact times. Seeley et al. [55] tested the accuracy of a nanocomposite piezoresponsive foam (NCPF) inserted into the running shoe under the insole, and comparisons with an instrumented treadmill revealed an error for predicted vGRF load rates between 22 and 29% for the NCPF. Finally, Seiberl et al. [63] analyzed the accuracy of Loadsol compared to a force plate during running and reported high precision of the sensors. However, the authors stated that insole devices are not accurate enough for highly dynamic GRF assessments, such as force rate.

Included in the data material are also three studies on running that have compared impact force assessments from accelerometry with force plate data. Ngho et al. [47] and Pogson et al. [49] both concluded that estimations of vGRF from accelerometer data using neural network modeling were promising, reporting small differences compared to force plate data. Sensors were placed on the trunk and on top of the shoe, respectively. Derie et al. [37] utilized machine learning to predict maximal vertical instantaneous loading rate (VILR) from triaxial accelerometers attached bilaterally to the tibia and concluded that multiple 3D tibial acceleration features gave a more accurate prediction of the VILR than the frequently used axial peak tibial acceleration, which is a single time discrete variable of tibial acceleration. Finally, two studies investigating landing impacts with validation from force plate data were identified, namely after somersault in gymnastics [31] and in ballet [62]. Whereas the first mentioned study concluded that external impact forces could not be estimated accurately based on accelerometer data, the second stated strong correlations between impact acceleration and peak vGRF.

The gold standard for GRF measurements are force platforms or treadmills instrumented with force sensors. However, such assessments are not available for the capture of complex movement patterns performed in the field, such as in team sports and downhill skiing. For running, an instrumented treadmill can provide the possibility to measure GRF from repeated foot strikes, simulating a long run. However, the environment will be controlled and thus not reflect variations posed by natural surfaces and terrain. Therefore, wearable technology could represent a considerable potential within this context. A systematic review from Ancillao et al. [93] found sensors that allow direct measurements of GRF—such as insoles, wearable load cells, or ad hoc designed pressure sensing devices—to be more reliable than GRF predicted from IMU data. This is confirmed in a recent review by Blazey et al. [87] who found instrumented insoles, in particular the Loadsol system, to offer a good in-field assessment tool. However, it is important to note that when sensors are worn under the foot, they compromise the foot–ground interaction, and the loads measured do not reflect the pressure absorbed by the tissue but rather the pressure on the device or the shoe to insole interface [87,93]. It is further emphasized that even if there is a correlation between predicted and directly measured GRF, it is difficult to estimate the absolute value of the peak force.

### 4.4. Which Knowledge Gaps Are Apparent in the Literature Regarding Wearable Sensor Technology for Assessments of Shock Impacts within Sports?

As previously discussed, an observed knowledge gap in this scoping review is the lack of studies assessing the magnitude of single shock impacts during running, landing, abrupt changes of direction, and pivoting activities in different invasion sports and other team sports. As monitoring in these sports is performed under the purpose of managing total strain of training and competition to prevent overtraining and injuries, knowledge on the magnitude of each single shock impact would add significantly to what can be drawn from the quantification of multiple impacts and exposure over time. Likewise, measurements of single shock impacts would also be of interest in the track and field events such as the high jump, long jump, and triple jump, but no studies could be found. Furthermore, it was somewhat surprising that no studies could be found on winter sports with known single shock impacts of considerable force, such as in alpine skiing (especially the downhill and Super-G disciplines), freeskiing, snowboard, and ski jumping. As described before, studies on these sports utilizing wearable technology primarily have focused on technique and tactical dispositions. To fully understand the consequences of shock impacts during real-life training and competition in various and often demanding environments, laboratory studies alone are not sufficient.

The aim of this scoping review was to summarize what has been done previously and carry out data charting, and not to score study quality. Still, it must be mentioned that many of the included studies are characterized by a relatively low number of participants. Overall, 18 of the included studies had one to 20 participants, seven studies had 21–50 participants, five studies had between 51 and 100, and three studies had more than 100 participants. One study did not inform about the number of participants at all. In the studies with more than 50 participants, three studies assessed head impacts in invasion sports and two studies assessed running. Finally, in studies with more than 100 participants, two assessed head impacts in invasion sports and one study assessed running. Furthermore, as many as 20 out of the 34 included studies were based on a single intervention and/or methodological study design, with validation of the wearable sensors utilized as a primary goal. All the included studies assessing shock impacts in running fall under this study design category. This reveals that this field still should be characterized as being immature, and that measurement methods for shock impacts during real-life running using wearable sensors are not yet adequate as a stand-alone approach. Of the remaining designs, nine studies were prospective cohort studies on invasion and team sports where participants were followed over one or more seasons. Eight of these studies investigated head impacts [36,40,42,45,48,51,54,58] and one study investigated whole body impacts [39]. In addition, two studies investigated head impacts in a laboratory using video recordings to verify impacts measured by the wearable sensors [41,53]. From these studies, it seems that the number of head impact events may be overestimated from sensor data alone. This is explained from difficulties in separating real game-related head impacts, which are defined as direct contact to the head from collisions or blows, to impacts not affecting the head, rapid changes in direction, landings, or other abrupt movements. Head impacts may further be overestimated with regard to severity when trusting sensor data alone. Valid assessments of head impacts during game play in invasion and team sports are still dependent on confirmation by video to identify true—and potentially damaging—impacts. In sum, the included studies show that also in this context, wearable sensor technology at this point does not provide the necessary accuracy as a stand-alone method.

Finally, the numerous brands and types of wearable sensors utilized in different studies may pose a challenge when it comes to comparing protocols, field set-ups, and results. Additionally, there are several pitfalls related to the before mentioned lack of consensus regarding placement of sensors, as well as signal processing and filtering, that can lead to inaccurate interpretation of data [85]. Technological development in wearable sensor technology is fast, with a multitude of suppliers. Five years ago, Sperlich and Holmberg [94] pointed out that studies regarding the validity and reliability of wearable sensors had shown that many of the sensor technologies available on the market had questionable validity and reliability when used in various sport settings and populations. This review indicates that this concern is still valid for the particular scope of shock impacts, as so many of the studies are addressing validation. Additionally, no studies aiming to compare or validate different types or brands of accelerometers or IMUs up against each other were identified. Thus, caution should be made when considering the usability of wearable sensors of this type for different sports and movement contexts, as it is unknown whether there may be differences in accuracy and validity between different brands. IMUs are marketed as valuable tools for coaches and athletes. However, the findings from this scoping review support previous concerns that wearable sensors alone at present is not adequate to ensure valid and accurate data on shock impacts in the field. There is a need for future validation studies including larger populations, taking into account the specific characteristics of defined sport-specific tasks. Furthermore, validation studies should in addition to comparison with gold standard technologies include comparisons of different types and brands of sensors, sensor placement, as well as filtering and cut-off values for the detection and definition of thresholds to separate true impacts from acceleration caused by rapid movements.

### 4.5. Limitations

To our knowledge, this is the first scoping review to address the use of wearable sensor technology for assessments of shock impacts within the domains of sports and occupational settings. A generic limitation of the scoping review methodology is that it does not allow for a formal evaluation of study quality and the level of evidence, as the included papers represent a wide array of study designs and methods [95]. This is valid also for this study. The scoping review approach was chosen with intent to provide an overview of the body of literature within the two defined contexts and to identify apparent knowledge gaps, which may guide future initiatives for research within this field [96]. Sampling frequency or filtering of data were not included in the research questions of this study. In retrospect, it seems relevant to suggest that selected sampling frequencies and filtering techniques, including cut-off values, could be further investigated in future studies. The validity of wearable sensors in different contexts depends not only on placement of the sensors but also on the methods of data analysis. From the studies included in this review, this seems particularly relevant for complex movements with rapid changes of directions. The use of wearable sensors, especially within the sports domain, is expanding fast. Thus, we cannot eliminate the possibility that new studies accommodating the inclusion criteria may have been published between the search for eligible studies and publication of this review. Finally, we cannot eliminate the possibility that relevant papers might have been left out, due to the definitions and choice of terms in our search strategy.

## 5. Conclusions

The main aim of this scoping review was to provide an overview of existent methods for assessments of shock impacts based on wearable sensor technology within two domains: sports and occupational settings. No studies were found on occupational settings, and this is the most prominent knowledge gap identified in this study. For the sports domain, accelerometry was the dominant type of wearable sensor technology utilized for assessing shock impacts, interpreting peak acceleration as a proxy for impact. Of the 34 studies included, 28 assessed foot strike in running, head impacts in invasion and team sports, or different forms of jump landings or plyometric movements. The methodology of included studies revealed that there is a lack of consensus regarding sensor placement and interpretation of the results. Furthermore, the included studies that aimed at validation up against established gold standard methods support previous concerns that wearable sensors alone at the present time are not adequate to ensure valid and accurate data on shock impacts in the field. This advocates for high-quality research being needed to find the appropriate sensors and methodology to utilize the potential of measuring shock impacts in the field by wearable sensor technology.

## Figures and Tables

**Figure 1 sensors-21-04962-f001:**
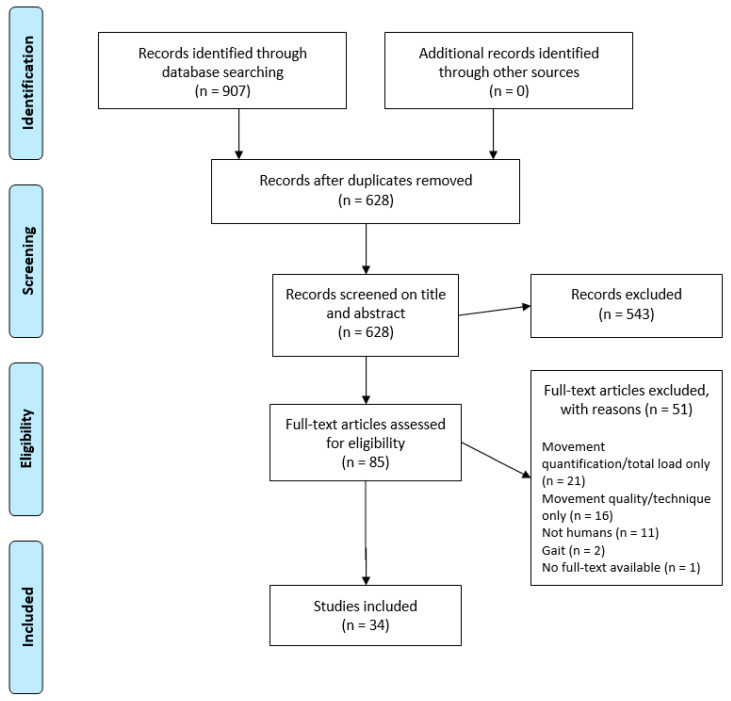
Flow diagram of selection of sources of evidence.

**Table 1 sensors-21-04962-t001:** Synthesis of results from included papers (*n* = 34).

Author(s) (Year)	Study Design	Study Aims	Main Outcome Measure of Relevance	Population (*n*, Mean Age ± SD)	Type(s) of Wearable Sensor Technology (Brand) and Placement of Sensor	Additional Technology for Validation	Key Findings of Relevance
**RUNNING**							
Ching et al. (2018)	Cross-over study, test–retest	To compare the impact loading during distracted running before and after a two-week auditory feedback gait retraining program.	Peak positive acceleration (PPA)	Male and female (7/9) recreational runners (*n* = 16, 25.1 ± 7.9 years)	Tri-axial accelerometer (Maestro WB, 01DB-Stell, Limonest, France). IMUs placed on the heel counter of the shoe and at the anteromedial right tibia.	Instrumented treadmill	Real-time auditory feedback gait retraining is effective in impact loading reduction during distracted running. Runners after gait training do not benefit from augmented auditory feedback.
Colapietro et al. (2019)	Observational single intervention case-control study	To evaluate spatiotemporal, kinematic, and kinetic measures during the loading response of running using a wearable sensor during two 1600 m track runs at different intensities between recreational runners with and without chronic ankle instability.	Horizontal component of change in acceleration of the foot at initial contact (breaking g), vertical component of change in acceleration of the foot at initial contact (impact g)	Male and female (8/10) recreational runners (*n* = 18, 22.7 ± 4.7 years)	RunScribe™ Pro 2x running sensor with integrated triaxial accelerometer and gyroscope (RunScribe Labs, Half Moon Bay, CA). IMU placed on the heel counter of the shoe.		Altered running mechanics were demonstrated in individuals with CAI compared to healthy runners. The clinical utility of wearable sensors in this context should be noted.
Derie et al. (2020)	Methodology study	To evaluate the performance of accelerometer-based predictions of the maximal vertical instantaneous loading rate (VILR) with various machine learning models trained on data of 93 rearfoot runners.	Axial peak tibial acceleration (APTA), vertical instantaneous loading rate (VILR)	Male and female (55/38) recreational runners (*n* = 93, 35.3 ± 10.0 years)	Tri-axial accelerometers (LIS331, Sparfkun, Colorado, United States). Accelerometer attached bilaterally to the anteromedial side of tibia.	Force plates	Applying machine learning to multiple 3D tibial acceleration features results in a more accurate prediction of the VILR than the frequently used APTA, which is a single time-discrete variable of tibial acceleration. A subject-dependent model resulted in the most accurate predictions compared to subject-independent models.
Gageler et al. (2013)	Single-intervention test	To explore the feasibility of using simple kinematic parameters obtained from a single inertial sensor to determine the rate of breaking and stopping, and to observe which limbs are most affected by the various rates of deceleration. Explore the contributions of each segment in absorbing the forces associated with breaking.	Peak accelerations and shock attenuation	Experienced runners, recreational and professional (*n* = 3, gender and age not given)	Custom-built tri-axel ±8 g accelerometer. Sensor positioned around the middle to upper thoracic vertebrae. Custom built device containing a 100 g accelerometer. Sensor attached bilaterally to the distal fibulas.	Motion capture system with eight cameras (OptiTrack)	Peak ankle impact increases when the rate of deceleration increases. The rate of deceleration did not alter the peak torso acceleration. It is difficult to detect the rate of stopping using only peak impact parameters from an inertial sensor unit located on the upper torso.
Mitschke et al. (2017)	Single intervention test/validation of method	To investigate the influence of different inertial sensor sampling frequencies on kinematic, spatiotemporal and kinetic parameters during rearfoot running.	Peak tibial acceleration and peak heel acceleration	Male recreational runners (*n* = 21, 28.9 ± 10.8 years)	Uniaxial light-weight accelerometer (ADXL78, Analog Devices), placed at the medial aspect and mid location between malleolus and plateau of the right tibia, and an IMU, combining a biaxial accelerometer (ADXL278, Analog Devices, and a biaxial gyroscope (IDG-650, InvenSense), affixed to the heel cup of the right shoe.	Force plate	When investigating peak heel acceleration or parameters which are directly derived from the accelerometer signal (e.g., touchdown), sampling frequency should be as high as possible or at a minimum of 500 Hz. 200 Hz were required to calculate parameters accurately for peak tibial acceleration.
Mitschke et al. (2018)	Single intervention test	To investigate the influence of the accelerometer operating ranges (OR) on the accuracy of stride length, running velocity, and on peak tibial acceleration when reducing the OR stepwise from ± 70 g to ± 32, ± 16, and ± 8 g.	Peak tibial acceleration (operating ranges, where g is the acceleration of gravity)	Male recreational runners (*n* = 21, 24.4 ± 4.2 years)	Individually configured IMU combining a biaxial accelerometer (ADLX278, Analog devices with OR ± 70 g and a biaxial gyroscope (IDG-650, InvenSense), with OR ± 2000 deg/s. IMU affixed to the heel cup of the right shoe. Uniaxial accelerometer (ADLX78, Analog devices) with OR ± 70 g. Accelerometer attached at the medial aspect mid-distance between the malleolus and the plateau of the right tibia.		Operating ranges influenced the outcomes of all investigated parameters. The lower ORs were associated with an underestimation error for all biomechanical parameters, which increased noticeably with a decreasing OR. Accelerometers with a minimum OR of ± 32 g should be used to avoid inaccurate measurements.
Ngho et al. 2018)	Single intervention study/validation of method	To investigate the use of neural network model (NN) and accelerometer to estimate vertical ground reaction force (VGRF).	Foot forward acceleration, segmented and normalized from foot initial contact to end contact (threshold at 5 N) and used as input for NN	Male subjects (*n* = 7, 21.3 ± 0.5 years)	Inertial sensor measuring acceleration, angular velocity, and magnetic field (Opal, APDM Inc.) Only forward acceleration is used for analyses; thus, it is referred to as uniaxial accelerometer. Sensor placed on top of the running shoe above the third metatarsal.	Force plates	Using NN and a uniaxial accelerometer simplified the estimation of VGRF, reduced the computational requirement, and reduced the necessity of multiple wearable sensors to obtain relevant parameters.
Pogson et al. (2020)	Single intervention study/validation of method	To present an artificial neural network method to predict ground reaction force (GRF) time series from a single, commonly used trunk-mounted accelerometer.	Estimated GRF	Physically active male and female (10/5) team sport players (*n* = 15, 23 ± 1 years)	GPS-embedded accelerometer (MinimaxX S5, Catapult Innovations, Scoresby, Australia). Accelerometer worn in a tight-fitting vest on the back of the upper torso.	Force plate	GRF was predicted with an average *r* 2 of around 0.9 for the time series of each impact, and the method therefore offers a promising approach to estimate GRF in the field.
Reenalda et al. (2016)	Single intervention test	To present a measurement set-up based on IMUs, to perform a continuous 3D kinematic analysis of running technique during the course of an actual marathon to objectify changes in running mechanics.	Peak center of mass (COM) vertical displacement and acceleration	Well-trained male distance runners (*n* = 3, 38.7 ± 8.2 years)	Eight inertial magnetic measurement units (MTw, Xsens Technologies B.V., Enschede, the Netherlands) containing a 3D accelerometer, a 3D gyroscope, and a magnetometer. IMUs placed on trunk (sternum, just below the sternal angle), pelvis (on the sacral bone between left and right iliac spine), upper legs (on tibial tract, halfway iliac crest, and lateral condyle of the tibia), lower legs (at the lower third of the medial surface of the tibia), and feet.	GPS enabled watch	Peak COM acceleration (derived from the sacral sensor) increased in all three runners and might indicate higher loading rates, a reduction in shock absorption quality, and a higher impact on the body. The presented measurement technique allows for more in-depth study of the running mechanics outside the laboratory and of the effects of fatigue on running mechanics.
Ruder et al. (2019)	Single intervention study	To examine the relationship between foot strike patterns (and impacts across a marathon race: (1) compare landing impacts quantified by tibial shock, between rearfoot, midfoot, and forefoot strike (RFS, MFS, and FFS, respectively), (2) examine the relationship between TS and speed across FSP, (3) to investigate the effect of fatigue on impacts.	Tibial shock (peak tibial acceleration)	Male and female (119/103) females marathon participants (*n* = 222, 44.1 ± 10.8 years)	Accelerometer device (IMeasureU BlueThunder IMU, Auckland, New Zealand). Sensor placed on the anteromedial aspect of their right distal tibia.	Video camera (to detect FSP before running the marathon)	Findings suggest that MFS runners exhibit similar impacts as RFS, and both exhibit higher impacts than FFS. RFS and MFS both exhibit increasing impacts with increasing speed, whereas FFS runners do not. RFS and MFS runners are similar in their impact loading. An FFS pattern may be protective against increasing impacts with increasing speeds.
Seeley et al. (2020)	Single intervention test/validation of method	To test the accuracy of a nanocomposite piezoresponsive foam (NCPF) that can be inserted into the running shoe under the insole in predicting important characteristics of vertical ground reaction force (vGRF) during running at three different speeds.	Impact peak vGRF, active peak vGRF, max impact rate	Male and female (17/14) recreational runners (*n* = 31, 23 ± 3 years)	Right shoe instrumented with NCPF sensors and an accelerometer (Bosch Sensortec, Mount Prospect, IL, USA). NCPF sensors placed under the insole (toe, ball, arch, heel), accelerometer attached to shoelaces.	High-speed video, force instrumented treadmill	Percent error was relatively low for predicted vGRF impulse (2–7%), active peak vGRF (3–7%), and ground contact time (3–6%), but relatively high for predicted vGRF load rates (22–29%). For each response variable of interest, the most accurate models were subject-specific models.
Seiberl et al. (2018)	Single intervention test/validation of method	To compare the accuracy and precision of a new wireless insole force sensor for quantifying running-related kinetic parameters over an extended period of use to a gold standard device in a laboratory setting.	vGRF parameters (ground contact time, area under the force–time curve, active peak force, time to active peak force, and both positive and negative force rate)	Male and female (4/6) (sport students (*n* = 10, 21.8 ± 0.8 years)	Ergonomic and linearly sensitive capacitor-based sensor; Loadsol^®^ insole (novel GmbH, Germany). Sensor placed inside shoes during running, under a tight-fitting sock for GRF comparative measurements.	Force plate	The mean bias of ground contact time, impulse, peak force, and time to peak ranged between 0.6% and 3.4%, demonstrating high accuracy of Loadsol^®^ devices for these parameters. 95% of all measurement differences between insole and force plate measurements were less than 12%. Highly dynamic behavior of GRF, such as force rate, is not yet sufficiently resolved by the insole devices, which is likely explained by the low sampling rate.
Van den Berghe et al. (2019)	Test–retest	Reliability and validity data for axial and resultant peak tibial acceleration (PTA) along the speed range of over-ground endurance running is lacking. The study developed a wearable system to continuously measure 3D tibial acceleration and to detect PTAs in real time.	Tibial acceleration, ground reaction forces and detected PTAs	Male and female (7/6) uninjured rearfoot runners (*n* = 13, 33 ± 13 years)	MEMS tri-axial accelerometers (LIS331, Sparkfun, Niwot, CO, USA). Sensor placed on a lower leg alongside the distal anteromedial aspect, 8 cm above the medial malleolus.	Instrumented treadmill with force plate	The wearable system developed was able to continuously detect PTAs in real time and can be used for applications aiming at monitoring (e.g., before, during, and after an in-field intervention) the impact loading experienced in the time domain by a runner during real world locomotion.
Wei et al. (2020)	Single intervention test	To explore whether running speed affects plantar load, and to compare plantar loads between habitual rearfoot strike (RFS) and non-RFS (NRFS) runners under their preferred running speed.	Plantar loads (*F*)	Male distance runners (*n* = 66, 25.2 ± 3.5 years)	Novel Pedar-X system (Novel, Munich, Germany), each insole including 99 force sensors. Insole placed in right shoe.	Photoelectric timing system, high-speed video camera	55% of participants were verified as habitual RFS and 45% were verified as habitual NRFS. Habitual runners tend to adjust their contact area according to the running speed through midfoot and forefoot regions. RFS runners remain susceptible to high impact force on the heel and midfoot, and NRFS runners experience high impact force in the first metatarsal regions.
Wouda et al. (2018)	Single intervention study/validation of method	To examine the validity of a method (using artificial neural networks) to estimate sagittal knee joint angles and vertical ground reaction forces (vGRF) during running using an ambulatory minimal body-worn sensor setup.	Peak vGRF	Male experienced runners (*n* = 8, 25.1 ± 5.2 years)	Xsens MVN Link inertial motion capture system consisting of 17 IMUs (Xsens, Enschede, the Netherlands). Full body Lycra suit used for placement at both shoulders, upper arms, lower arms, hands, upper legs, lower legs, feet, head, sternum, and pelvis.	Motion capture system with high-speed cameras and instrumented treadmill with force plate	Sagittal knee kinematics and vGRF can be estimated using only three inertial sensors placed on the lower legs and pelvis. The peak vGRF are estimated with no significant differences with respect to the reference. Best performance can be obtained when the proposed approach is applied to a single subject.
**INVASION AND TEAM CONTACT SPORTS (activity specified in first column)**							
Mihalik et al. (2016) *Football*	Prospective cohort study (throughout seasons over 8 years)	To investigate the clinical utility of head impact magnitude thresholds used by various commercially available head impact indicators to positively predict concussion among American football players.	Head impacts (impacts exceeding 10 g of peak linear acceleration)	Division I Football Bowl Subdivision college football players (*n* = 185, 19.2 ± 1.4 years)	Head Impact Telemetry (HIT) System; including linear acceleration, rotational acceleration, HIT severity profile (HITsp), head injury criterion (HIC), and Gadd severity index (Riddell Corp., Chicago, IL, USA). Sensors embedded in helmet.		The ability of a head impact indicator—used in isolation—to detect a concussive injury is minimal, even if it can accurately measure and report biomechanical outcomes. Injury thresholds used by existing head impact indicators cannot predict concussion when used in isolation.
Rose et al. (2018) *Football*	Longitudinal cohort study (throughout season)	To determine the association of repetitive subconcussive head impacts with functional outcomes in primary and high school tackle football players.	Head impacts ≥ 10 g	Primary school and high school football players (*n* = 112, 13.6 ± 2.9 years)	Riddell SpeedFlex or Speed helmet (Riddell, Rosemont, Illinois), containing the Riddell InSite Impact Response System, based on the Head Impact Telemetry (HIT) system Sensor placed between the player’s head and the helmet padding.		In youth tackle football, subconcussive head impacts sustained over the course of a single season may not be associated with neurocognitive functional outcomes. The absence of a significant association may reflect the relatively short follow-up interval and signals the need for studies across multiple seasons.
McIntosh et al. (2018) *Football*	Laboratory tests and prospective observational study (over two games)	To assess the utility, functionality, and wearability of the X-Patch^®^ as a measurement tool to study head impact exposure in sports without helmets, using Australian Football as an exemplar sport.	Head impacts (peak linear acceleration ≥ 10 g).	Male and female (24/29) amateur level adult Australian football players (recruited *n* = 97, included in analysis *n* = 53, 26.0 ± 2.0 years)	X-Patch (X2 Biosystems, Seattle, WA, USA). Sensor placed over the mastoid process.	Video analysis	The X-Patch had limitations in two distinct areas: accuracy (laboratory tests) and validity (field tests). The potential errors were considerable and could result in sizable misreporting of the head impact incidence rates. Use of the current X-Patch^®^ devices should be limited to research only and in conjunction with video analysis.
Muise et al. (2016) *Football*	Observational study (throughout season)	To examine how frequently, and to what magnitude, Canadian University football players get hit in training camp and how this compares to practices and games in regular season.	Head accelerations greater than 15 g	Male players in Canadian Interuniversity Sport (CIS) football (*n* = 47, age not given)	GForce Trackers (GFT) (Artaflex, Markham, Ontario, Canada). Sensor placed inside football helmet, to the right side of the crown.		Data from 20,950 impacts revealed that games were associated with significantly larger magnitudes and frequencies than either training camp or practices, but that training camp was associated with significantly greater magnitudes and frequencies than in-season practices. In addition, positional differences existed.
Bruce et al. (2019) *Basketball*	Single intervention study	To quantify the influence of basketball court surface construction and shoe midsole stiffness on ground reaction force (GRF), lower-extremity joint work, impact, and impact attenuation during countermovement jump landings.	Resultant peak acceleration tibia and head impact attenuation	Male collegiate and high school basketball players (*n* = 29, 19.1 ± 3.3 years)	Uniaxial and tri-axial piezoelectric accelerometers (ADXL78 and ADXL1002, Analog Devices, Inc., Norwood, MA, USA). ADXL78 mounted to the forehead and ADXL1002 mounted to the tibial tuberosity.	Force plates, high-speed cameras and a Vertec Jump trainer^TM^	Shoe stiffness and surface had minimal effects on parameters associated with impact during countermovement jump landings. Landing in a more compliant shoe reduced peak ankle moment and tibial impact acceleration. Results for tibial impact acceleration were inconsistent.
Cortes et al. (2017) *Lacrosse*	Prospective cohort study (throughout season)	To utilize video analysis to verify head impact events recorded by wearable sensors and describe the respective frequency and magnitude.	Head impacts with linear acceleration ≥ 20 g; leading to recording of impact	Male and female (30/35) high school lacrosse players (*n* = 65, 16.4 ± 1.3 years)	Females: X-patch sensors including a triaxial accelerometer and a gyroscope (X2 Biosystems, Seattle, WA, USA). Sensor placed at the right mastoid process. Males: GForce Tracker including a tri-axial accelerometer (GForceTracker Inc.). Sensor affixed to the inside crown of the helmet.	Video recordings	65% and 32% of all head impacts recorded during boys’ and girls’ lacrosse game play were verified as true game play-related head impacts by video analysis, respectively. Results suggest that existing wearable sensor technologies may substantially overestimate head impact events.
Kelshaw et al. (2018) *Lacrosse*	Prospective cohort study (throughout season)	To assess the effects of isometric cervical muscle strength (ICMS) on head impact kinematics (HIK) in high school boys’ lacrosse, and to investigate the relationship between cervical anthropometric measures (CAM) and ICMS.	Head impacts (linear acceleration (g) and rotational velocity in degrees per second)	Male high school varsity lacrosse players (*n* = 15, 16.5 ± 1.33 years)	IMU with tri-axial accelerometer and gyroscope (GForceTracker™, Markham, ON, Canada). Sensor adhered to the inside crown of the helmet.	Handheld dynamometer to measure ICMS, high-definition camera	Thirteen of the participants sustained game-related impacts that were confirmed on video. A total of 367 impacts were confirmed using video analysis for the 13 participants. ICMS did not affect HIK, and CAM did not approximate ICMS. Findings suggest that greater ICMS alone may not mitigate HIK in collision sports.
Vollavahn et al. (2018) *Lacrosse*	Prospective cohort study (throughout season)	To establish the frequency of head impacts across impact mechanism, and to determine differences in linear and rotational head impact accelerations according to impact mechanism in NCAA Division III men’s lacrosse athletes.	Head impacts, threshold 10 g of linear acceleration	Male athletes in NCAA Division III men’s lacrosse (*n* = 11, 20.9 ± 1.3 years)	xPatch sensors (X2 Biosystems, Seattle, WA, USA) containing a triaxial accelerometer and rotational gyroscope. Sensor placed over the participants’ right mastoid process.	Video camera	A total of 167 head impacts were successfully verified and coded with a mechanism using video footage during 542 total participant exposures. The highest incidence rate was head to body, and the lowest was head to ball. The study failed to find differences in head impact magnitude depending on the mechanism that caused the impact.
Grainger et al. (2018) *Rugby*	Longitudinal observational cohort study (throughout season)	To compare the absolute and relative number of impacts between nine positional groups in rugby union.	Impacts defined from values above 2 g in a 0.1 s period (of which 10 g impact classifications are likely to be accrued by collisions)	Professional male rugby players (*n* = 38, 26.4 ± 4.7 years)	GPS units with integrated triaxial accelerometers (StatSports Viper, Northern Ireland). Incorporated in jerseys, on the thoracic spine between the scapulae.		The frequency and magnitude of impacts experienced by positional groups vary. Inertial sensor impacts encountered during match play are likely a combination of “real physical impacts” from collisions and those accrued from movement tasks (deceleration, landings, and changes of direction). It is important to assess the total inertial sensor impact values accrued during match play with caution.
Patton et al. (2020) *Soccer*	Prospective cohort study (over two seasons)	To (1) identify the percentage of video-confirmed events recorded by headband-mounted sensors through video analysis; (2) compare video-confirmed events with the classification by the manufacturer filtering algorithms; and (3) quantify and compare the kinematics of true- and false-positive events.	Resultant linear acceleration and angular velocity	Male and female (49/23) adolescent varsity soccer players (*n* = 72, age not given)	Triaxial gyroscope for measurement of angular velocity and a high- and low-g triaxial accelerometer for measurement of linear acceleration (SIM-G), trigger threshold 16 g. Sensor mounted in a neoprene headband and positioned just above the greater occipital protuberance.	High-definition video camera	Of the 1893 sensor-recorded events in the final dataset, video confirmation revealed that 1316 (70%) were impact events, 396 (21%) were trivial events, and 181 (10%) were non-events. Percentages of video-confirmed impact events, trivial events, and non-events varied by sex. Current manufacturer filtering algorithms and magnitude thresholds are ineffective at correctly classifying sensor-recorded events.
Sandmo et al. (2019) *Soccer*	Descriptive laboratory study/single intervention study	To test the validity of an in-ear sensor for quantifying head impacts in youth soccer.	Head impacts (peak linear acceleration (PLA), peak rotational acceleration (PRA), and peak rotational velocity (PRV))	Male youth soccer players at the regional elite level (*n* = 6, 15.3 ± 0.3 years)	Sensor device MV1 (MVTrak). Placement of sensor in the left external ear canal.	Digital video cameras	The in-ear sensor displayed considerable random error and substantially overestimated head impact exposure. Despite the sensor’s excellent on-field accuracy for discriminating headings from other accelerative events, there is a need for secondary means of verification (e.g., video analysis) in real-life settings.
Saunders et al. (2020) *Soccer*	Prospective cohort study (throughout season)	To compare head impact magnitude and frequency between men’s and women’s intercollegiate soccer players based on head impact mechanism.	Head impacts ≥ 10 g (linear acceleration (g) and rotational acceleration (deg/s2)	Male and female (12/16) intercollegiate soccer players (*n* = 28, 20.1 ± 1.1 years)	xPatch (X2 Biosystems, Seattle, WA, USA) head impact sensors. Contains a triaxial accelerometer and gyroscope. Sensor placed over the participants’ right mastoid process.	Video camera	Only head impacts that could be clearly seen on video were included. Most of the head impacts recorded in the current study were below 20 g. Men’s soccer athletes sustain head impacts more frequently than women. Women had the highest head impact frequency when heading a soccer ball, while men were most likely to sustain head to body contact.
**PLYOMETRIC ACTIVITIES AND LANDINGS (activity specified in first column)**							
Nagano et al. (2018) *Badminton*	Observational study (two games)	To elucidate the movements requiring greater trunk accelerations and its frequencies during badminton games and compare the acceleration components among such movements.	Resultant trunk acceleration >4 g	Female badminton players (*n* = 10, 15.8 ± 1.0 years)	Triaxial accelerometer (SS-WS1201, Sports Sensing, Fukuoka, Japan). Accelerometer secured to the upper back using a vest.	Two digital video cameras	The movements generating greater trunk acceleration were running during an underhand stroke on the dominant hand side leg, landing after an overhand stroke on the dominant and non-dominant hand side legs, and cutting from a split step using the dominant and non-dominant hand side legs.
Almonroeder et al. (2019) *Ballet*	Single intervention study/validation of method	To compare impact forces in ballet measured by a wearable accelerometer to loading rates of GRF during a common ballet maneuver involving jumping.	Peak impact acceleration and peak vertical GRF	Experienced female ballet dancers (*n* = 15, 18 ± 4.5 years)	Tri-axial accelerometer/IMU (Shimmer3 IMU). Sensor placed on the left anterior superior iliac spine, with the vertical axis of the device aligned with the long axis of the trunk when standing.	Force plates	Strong positive correlations were found between impact accelerations and peak vertical GRF, indicating that a wearable accelerometer can provide a means of indirectly capturing GRF features in ballet dancers during landing.
Bradshaw et al. (2020) *Gymnastics*	Single intervention study/validation of method	To examine measurement agreement between resultant peak force with vertical peak force and peak resultant deceleration of backward somersault landings.	Peak resultant and vertical ground reaction force (GRF), peak resultant deceleration	Competing female artistic gymnasts (*n* = 7, 10–15 (± not given) years)	Iso-inertial measurement unit/ IMU (iMeasureU, Auckland, New Zealand) IMU placed on upper back (T2).	Force plates	Moderate measurement agreement was revealed between the peak resultant force and peak deceleration measures, indicating that the external impact forces for backward somersault landings cannot be adequately estimated using an IMU/accelerometer placed on the upper back. Alternative placement of IMUs needs to be explored (ex. distal tibia, L5).
Campbell et al. (2020) *Gymnastics*	Single intervention study/validation of method	To determine the most appropriate filter cut-off for acceleration- and force–time data when measuring peak resultant acceleration (PRA) and ground reaction force (PRGRF) during landings after standing backward handspring and backward somersault from a height.	Peak resultant acceleration	Male and female (8/8) competitive artistic gymnasts (*n* = 16, 14.1 ± 3.6 years)	Triaxial IMU (iMeasureU, Auckland, New Zealand) IMUs placed on upper back (T2), lower back, both sides of the posterior superior iliac spine and forearms for backward handspring; on upper and lower back and bilaterally in tibia for backward somersault.	Force plate	For applied sports settings, no filtering is needed. However, a minimum cut-off of 85 Hz should be implemented for research purposes investigating pooled data from a large number of participants.
Stöggl et al. (2017) *Multiple motions*	Single intervention test/validation of method	To validate the accuracy of the OpenGo sensor insole, compared to the AMTI force plate system and the gold-standard sensor insole system, PedarX, during walking, running, jumping, body balance, and imitation motions.	Impulse of force, maximal force and mean force (g-range ± 8 g)	Male and female (14/2) sport science students (*n* = 16, 31 ± 10 years)	Sensor insoles OpenGo (Moticon GmbH, Munich, Germany). Consists of two sensor insoles (containing 13 capacitive sensors each) that measure the plantar pressure distribution and the acceleration in three dimensions in space. PedarX Mobile System (Novel GmbH, Munich, Germany) measures pressure distribution and used for comparison. Both insoles sandwiched between the foot and the inside of the shoes.	Force plate	The OpenGo system demonstrated almost perfect agreement with force plate data for detection of cycle characteristics and temporal parameters during gait and jumping tasks. Force impulses were 13–34% lower with OpenGo when compared to AMTI. During fast motions, with high force and impact, OpenGo provided lower force and latency in force kinetics. Thus, very short ground contact times with force impacts cannot be determined accurately by the OpenGo system.
Ziebart et al. (2017) *Multiple motions*	Single intervention study/validation of method	To examine how system characteristics, such as operating range and sampling rate, influence the measurement of peak impact loads by commercial activity monitoring systems as compared to a laboratory-grade criterion standard accelerometer.	Peak impact loads (acceleration gmax)	Male and female (5/7) (*n* = 12, 24.1 ± 2.6 years)	Three tri-axial accelerometers; a criterion standard laboratory-grade unit (Endevco 7267A) and two systems primarily used for activity monitoring (ActiGraph GT3X+, GCDC X6-2mini). All three accelerometers were affixed to one another using double-sided tape, secured to the participant’s left anterior superior iliac crest (ASIS).		Accelerometers designed for activity monitoring underestimated peak impact magnitude by up to 35%. Underestimation error was greater for tasks with greater impact magnitudes. Both the type and intensity of activity should be considered when selecting an accelerometer for characterizing impact events. In addition, caution may be warranted when comparing impact magnitudes from studies that use different accelerometers.
Ross et al. (2016) Conference abstract *Snowboard*	Not given	To report on the progress of a four-stage research program to develop an athlete tracking system suitable for use by snowboard athletes, where stage 2 investigates the relationship between body mounted accelerometers and landing impacts.	Not given	Not given	IMUs containing an accelerometer, gyroscope, and magnetometer sensor (The OptimEye (Catapult, Australia) and IMeasureU (I Measure U, New Zealand)). Sensor placement not given.	Force plate	Both IMUs showed similar correspondence with the gold standard (force plate). Findings will be used to refine data collection and processing techniques for stage four of the program; implementation and validation of an on-snow athlete tracking system.

**Table 2 sensors-21-04962-t002:** Excerpt of demographic and technological characteristics of all included studies (*n* = 34).

**Country of Origin ***	US (10), Canada (5), Australia (4), UK (4) Germany (3), Belgium (3), The Netherlands (2), Austria, China, Hong Kong, Japan, Malaysia, New Zealand, and Norway (all 1)
**Study Design**	Single intervention test/validation of method/methodology studies (20), prospective/longitudinal/observational studies (11), test–retest (2), not given (1)
**Participants** **(a)** **Number** **(b)** **Mean Age** **(c)** **Level**	(a) 1–10 participants (8), 11–20 participants (10), 21–30 (4), 31–40 (2), 41–50 (1), 51–100 (5), <100 (3), not given (1) (b) 10–15 years (5), 16–20 years (6), 21–30 years (13), 31–40 years (5), 41–50 years (1), not given (4) (c) Recreational/amateur (10), experienced/competitive (19), professional (1), not given (4)
**Type of Wearable Sensor Technology and Sensor Placement ***	Accelerometer/IMU on lower limb beneath the knee (11), accelerometer/IMU embedded in helmet or placed on the head (10), accelerometer/IMU multiple body segments—trunk, pelvis, body suit (9), insole force sensors (4), not given (1)
**Additional Technology for Validation ***	Force plate/instrumented treadmill (15), camera/video recordings (14), GPS enabled watch (1), handheld dynamometer (1), Vertec Jump Trainer (1), photo-electric timing system (1)

* Several studies have split origin between two countries and have utilized more than one type of wearable sensor and/or additional technology; thus, numbers may sum up to <34.

## Data Availability

No new data were created or analyzed in this study. Data sharing is not applicable to this article.

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
