# Peer review of "The Use of Wearable Sensor Technology to Detect Shock Impacts in Sports and Occupational Settings: A Scoping Review"

_sensors, 2021, doi:10.3390/s21154962_

Round 1
Reviewer 1 Report
Shock impacts in daily life activities can cause damage to the body. Therefore, it would be important to have thresholds for tolerable impacts and there is a need for methods that can accurately measure shock impacts in real-life settings. Accordingly, the aim of the authors was to present an overview (scoping review) of existing methods for shock impact assessments using wearable sensors in sports and demanding work environments. The authors included and analyzed 34 papers in their scoping review. No studies were found in the context of demanding work environments. The results showed that there is a lack of consensus regarding sensor placement as well as interpretation of the results. In addition, the authors found a high amount of validation studies within the included studies indicating that wearable sensors at present are inadequate as a stand-alone method for valid and accurate analysis of shock impacts in sports environments.
The authors have presented a relevant paper that fits the range of topics covered by the journal. The research question and the decision to conduct a scoping review are justified. The methodological procedure is described in detail, is comprehensible and seems adequate. The discussion is easy to follow, as the authors are explicitly guided by the questions introduced in the introduction. Finally, the authors also address limitations of their own work to give a balanced view of their work. In sum, the paper is clearly structured and easy to follow. It provides a good overview of the current state of research in the context of wearable sensor measurements for shock impacts in sports. Furthermore, research gaps are identified and recommendations for future research are derived.
Congratulations to the authors for a very good paper.
Author Response
We highly appreciate the time and effort the reviewer spent on our manuscript - thank you!
Reviewer 2 Report
This manuscript provides a review of wearable technologies and the measurements used to quantify shock/impact in sporting or demanding work environments. Please see some specific comments below:
- I think the paper title could be re-phrased. For example, page 2, line 47 would work as a better title for the paper.
- The authors should re-word parts of the report that are written in the first person. (i.e remove and re-word sentences containing "we").
- Some parts of the introduction could be made more concise as they detract from the main purpose of the review. For example, page 2, lines 51-66.
- The authors should clearly define what they mean by the term "demanding work environment" especially as this is included as inclusion criteria in the review process. Could the lack of definition have contributed to the lack of papers fitting that criteria?
- There are some papers which I would have expected to see in the review. Could the authors provide a comment as to why papers such as Gruber, A, Boyer, K, Derrick, T & Hamill, J. Impact shock frequency components and attenuation in rearfoot and forefoot running, Journal of Sport and Health Science, Volume 3, Issue 2, 2014, Pages 113-121 were not included as this paper would appear to meet all inclusion criteria?
- Some of the columns in Table 1 seem to repeat information and I would encourage the authors to reduce the content. For example, the content in Impact Site is already stated in the Study Aims column. The Study Aims and Key Findings column could certainly be made more concise. The order of the content in Table 1 could also be addressed. I think it would be better to order the content based on some of the key questions e.g. either sensor placements or types of activities. This would help readers compare and contrast studies more easily.
- Figure 2 and 3 content appear to be repeated in the text and I don't think they add anything to the review.
- Based on the review of the literature could the authors include some comments about what would be the most appropriate technology for different impact/shock conditions?
Author Response
We highly appreciate the time and effort the reviewer have spent reading our manuscript, and find the comments and suggestions to be of definitive value. We have tried to accommodate the suggested alterations in the revised manuscript. A detailed response can be found in the attached file "Response to reviewer 2_Eitzen et al".
